# In Situ Silver Nanonets for Flexible Stretchable Electrodes

**DOI:** 10.3390/ijms24119319

**Published:** 2023-05-26

**Authors:** Qingwei Liao, Wei Si, Jingxin Zhang, Hanchen Sun, Lei Qin

**Affiliations:** 1Key Laboratory of Sensors, Beijing Information Science & Technology University, Beijing 100192, China; siwei_2021@163.com (W.S.); 18554218027@163.com (J.Z.); sunhanchen_2001@163.com (H.S.); 2Key Laboratory of Modern Measurement & Control Technology, Ministry of Education, Beijing Information Science & Technology University, Beijing 100192, China; 3Key Laboratory of Photoelectric Testing Technology, Beijing Information Science & Technology University, Beijing 100192, China

**Keywords:** silver nanonets, silver nanowires, conductive materials, flexible stretchable material

## Abstract

Shape-controlled synthesis is an effective method for controlling the physicochemical properties of nanomaterials, especially single-crystal nanomaterials, but it is difficult to control the morphology of single-crystal metallic nanomaterials. Silver nanowires (AgNWs) are regarded as key materials for the new generation of human–computer interaction, which can be applied in large-scale flexible and foldable devices, large-size touch screens, transparent LED films, photovoltaic cells, etc. When used on a large scale, the junction resistance will be generated at the overlap between AgNWs and the conductivity will decrease. When stretched, the overlap of AgNWs will be easily disconnected, which will lead to a decrease in electrical conductivity or even system failure. We propose that in situ silver nanonets (AgNNs) can solve the above two problems. The AgNNs exhibited excellent electrical conductivity (0.15 Ω∙sq^−1^, which was 0.2 Ω∙sq^−1^ lower than the 0.35 Ω∙sq^−1^ square resistance of AgNWs) and extensibility (the theoretical tensile rate was 53%). In addition to applications in flexible stretchable sensing and display industries, they also have the potential to be used as plasmonic materials in molecular recognition, catalysis, biomedicine and other fields.

## 1. Introduction

Driven by the new generation of information and communication network technology, the manufacturing industry will rapidly complete the transformation of digitalization, networking and intelligence, of which the key source technology is perception. Soft perception is the key technical support for making perception free from multi-dimensional space. Flexible sensing can subversively change the rigid physical form of traditional information devices and systems and realize the efficient integration of information with people, objects and the environment. The flexibility of information acquisition, processing, transmission can be realized in the fields of the Internet of Things [1], portable wearable devices [2,3], environmental and medical monitoring and care [4,5,6,7,8], communication systems [9], human–computer interaction [10,11,12], AI “nurse” and on-body AI hardware [13], etc., so as to maximize the “interconnection of everything”. Among them, multi-dimensional spatial perception means that it can maintain its original function after repeated stretching, compression, folding, twisting and other morphological changes, and can be highly efficient and integrated with the perceived information so as to realize the multi-dimensional and high truth representation of the perceived information. As the core component of flexible electronic devices, flexible electrodes are also required to maintain good mechanical properties and stable electrical properties under external forces such as tension, bending and torsion within a certain range [14,15,16]. Therefore, the research on flexible stretchable electrode materials with extensibility, flexibility and high conductivity has become a hot issue in the fields of flexible sensing and flexible display [17,18,19,20,21].

At present, flexible stretchable electrodes are mainly realized by changing the electrode structure or adding conductive materials to the elastic substrates. The former is limited by the inherent properties of materials and has significant limitations, whereas the research on the latter is more open. Common conductive materials include ITO, silver nanowires (AgNWs), carbon-based composites [22,23], graphene [24,25], conductive polymers [26,27,28], etc. Among them, AgNWs have excellent conductivity, light transmission, bendability, stability and thermal conductivity, with a low production cost and no moiré. When applied to rough surfaces, the high anisotropy of AgNWs also makes it have certain advantages in forming infiltration networks. The preparation of AgNWs-based stretchable electrodes or interconnects on flat or wavy patterned PDMS substrates by spin coating [29], inkjet printing [30], spray coating [31] and brush painting [32] has also been extensively investigated and is considered as one of the electrode materials that can replace traditional indium tin oxide (ITO) for the new generation of flexible devices [33]. The morphology of AgNWs can be controlled by controlling the preparation process factors (temperature, reaction time, surface covering agent, ion concentration, etc.) [34]. Generally, the greater the aspect ratio of AgNWs, the better the conductivity and extensibility. Although AgNWs with a greater aspect ratio can be further prepared by electrospinning technology, the conductivity of flexible electrodes was reduced due to the poor fusion welding effect and high junction resistance of AgNWs. At present, nano welding and nano junction fastening technologies such as photonic sintering, mechanical pressing, gold plating, electrodeposition, plasma treatment, etc., are usually used to perform secondary (or multiple) processing on AgNWs to increase the contact area at the intersection and reduce the junction resistance so as to improve the charge transport capability, mechanical stability and integration of the conductive network [35,36,37]. At present, our group successfully synthesized shape-controlled AgNWs by improving the diol process and achieved the regulation of the diameter, length and size uniformity of AgNWs by studying the reaction temperature, different control agents and the drop-in rate of the AgNO_3_ solution [36]. In addition, flexible conductive fabrics with a high electrical conductivity were prepared by growing silver nanosheets on fabrics using an in situ growth process [38]. So far, most of the anisotropic nanocrystals prepared by morphology control are formed due to preferential growth in a single direction and are highly symmetrical. To increase the shape diversity of nanocrystals, seed overgrowth has been proven to be a general way to make the growth process enter other anisotropic modes and form nanocrystals with a complex morphology [39].

In this paper, silver nanonets (AgNNs) were designed and prepared, anisotropic polyhedral particles were obtained by etching silver nanoparticles (AgNPs) and then AgNWs were grown in different sections of the polyhedral particles and between different polyhedral particles by the AgNWs growth process so that AgNNs are grown in situ. Three-dimensional network nano silver conductive materials with high conductivity and extensibility were obtained, and the ultimate tensile rate of the mesh with different shapes was calculated. Due to the construction of the network structure, the conductive material could deform when the flexible device was stressed, and maintain stable conductivity within the ultimate tensile rate. In addition, the absorptivity of the AgNNs between 350 nm and 700 nm in the ultraviolet–visible (Uv-Vis) and near-infrared range between 800 nm and 1300 nm was also measured. It was found that the AgNNs have a strong plasmon resonance effect in the near-infrared band, which indicates that it has a wide application potential in near-infrared band Raman spectroscopy identification, catalysis, sensing and medicine. Finally, through the rod coating method and chemical impregnation method, the AgNNs were attached to PDMS, paper-based, non-woven fabric and polyester base materials with different weaving processes to prepare flexible stretchable conductive films and conductive fabrics. Its conductivity and extensibility are better than the conductive materials prepared by AgNWs, which provide a new idea for the design and preparation of nano silver conductive materials.

## 2. Results and Discussion

### 2.1. Extensibility of AgNNs

The high flexibility and toughness of AgNWs depends on the aspect ratio of AgNWs. Generally, the greater the aspect ratio of AgNWs, the better the conductivity and extensibility. Although AgNWs obtained by rod coating, spraying and other methods can also present a random distribution network, a considerable part of AgNWs will not contribute to charge transfer. Under the ultimate tensile condition of the substrate, the density per unit area is greatly reduced due to the absence of connections between the AgNWs, which will also lead to a decline in the photoelectric performance. As shown in Figure 1, multiple overlaps between AgNWs can be seen in the SEM image of the conductive material based on AgNWs, and the overlaps generate junctional resistance, making it less conductive. In contrast, the AgNNs were in the shape of a porous skeleton, and the AgNWs were connected with each other to form a conductive three-dimensional network, which increased the conductivity. AgNWs were grown together through the polyhedral particles, so no welding treatment was required for the contact points. Under the tensile condition, the overlap of AgNWs will be easily disconnected, which will lead to a decrease in electrical conductivity or even system failure. In contrast, AgNNs can be deformed as the substrate is stretched until the ultimate stretching rate of AgNNs is reached and the mesh changes to a wire shape. In addition, AgNNs can deform with the stretching of the substrate. When the substrate is stretched, the mesh shrinks and gradually becomes linear. The mesh structure is more closely connected due to the tensile action, so the conductivity will not decrease due to the increase in stress within the ultimate tensile rate.

In order to evaluate the tensile rate of AgNNs, different types of two-dimensional meshes were simultaneously stretched to one-dimensional linear meshes when the substrate material was stretched. Comparing the mesh tensile length with the original mesh diameter, the ultimate tensile rate of different types of mesh can be calculated. As shown in Figure 2, assuming that the average length of AgNWs formed between AgNPs in AgNNs is k nm, when the mesh is triangular, quadrilateral, pentagonal, hexagonal, heptagonal or octagonal, the tensile rate (δ) is expressed by Formula (1):(1)$$\delta =\left(L-R\right)/R$$
where L is the length of the mesh after it has been stretched and R is the original diameter of the different types of meshes.

Through calculation, the theoretical ultimate tensile rates for triangular, quadrilateral, pentagonal, hexagonal, heptagonal and octagonal lattices are 30.4%, 41.8%, 47.1%, 50%, 52.2%, and 52.7%, respectively. As the number of mesh sides increases, the inner angle increases, and the tensile rate increases gradually, tending to 53%. This indicates that when the tensile rate is less than 53%, the stress has little effect on the network connection growth point, its conductive three-dimensional network will not be destroyed and the conductivity will not decrease.

### 2.2. Formation Mechanism of AgNNs

#### 2.2.1. Etching of AgNPs

Providing an anisotropic growth site for AgNWs is the key for the preparation of AgNNs. The growth mechanism of AgNWs on the polyhedral particles is shown in Figure 3a. Acetone can be used as a cleaning agent to effectively remove impurities and oxides from the surface of AgNPs due to its good fat solubility and water solubility [40]. Br^−^ can etch AgNPs, changing their surface, morphology and optical properties, increasing the surface activity of AgNPs and enabling stronger interactions between silver particles and certain adsorbates. Wu’s team [41] found that the oxidative etching effect of Br^−^ increases with their concentration, while Br^−^ can combine with the free Ag^+^ in the initial stage, preventing the solution from being rapidly supersaturated by Ag seeds. As shown in Figure 3b–d, after etching the treated AgNPs using Br^−^, the relatively flat isotropic treated AgNPs has obvious edges and corners on the surface, which are transformed into anisotropic polyhedral particles. At the same time, the presence of Br^−^ and Cl^−^ in the system slows down the reduction of Ag^+^ and prevents the aggregation of silver nuclei. Ultimately, this enables the silver ions to grow in situ into AgNWs on one or more surfaces of the silver polyhedral particles.

#### 2.2.2. Formation of AgNNs

The formation of AgNNs in the diol system is divided into three main stages, as shown in Figure 4. First, as the temperature increases, the diol solution undergoes a chemical reaction in which the oxidized acetaldehyde continuously reduces silver ions from silver nitrate and the number of crystal nuclei in the system continues to increase. When the concentration of silver atoms reaches the saturation value, it starts to grow into a heterogeneous nucleus. In the second stage, the heterogeneous nucleus is deposited onto anisotropic polyhedral particles using the Ostwald maturation mechanism, as the smaller heterogeneous nucleus has a much higher surface energy than the polyhedral particles with diameters of 700–900 nm after etching (Figure 3b–d), which have a higher solubility and lower stability [42,43]. When they grow to a critical size, they form multi-helical particles that attach to the surface of the polyhedral particles. In the third stage, the coating agent PVP and the control agent (KBr and AgCl) are effectively combined with Ag to passivate the multi-helical particles of the silver {100} plane to form multiple twins and inhibit its lateral growth while promoting vertical growth. With the continuous addition of silver ions, the anisotropic growth of Ag atoms on the (111) crystal plane is active, and multiple twins are grown into AgNWs, realizing the in situ growth between polyhedral particles and different polyhedral particles, finally forming AgNNs.

After adding treated AgNPs to the reaction system, only a small amount of AgNWs grew in situ on polyhedral particles, and no network structure was formed. The increased size of the polyhedral particles produced by the etching of the treated AgNPs gives them a higher stability and lower solubility and therefore requires a longer reaction time to react with other substances [44]. Therefore, after the silver nitrate solution was completely dropped, we took out part of the AgNNs solution after stirring at 160 °C for 3 h, 12 h and 24 h, respectively, and centrifuged and washed it to investigate the effect of reaction time on the formation of AgNNs. When the reaction time was extended to 24 h, a large number of AgNWs (the diameter of AgNWs was 100–120 nm and the length was 4–6 μm) were grown in situ on polyhedral particles and between polyhedral particles, which formed a three-dimensional network structure of mesh distribution (as shown in Figure 5a,b).

To verify the structure and preferred growth direction of the AgNNs, the AgNNs were tested by XRD. As shown in Figure 5c, there are three distinct peaks in the XRD pattern of AgNNs at 38.1°, 44.3° and 64.4°, corresponding to the (111), (200) and (220) planes of Ag in JCPDS card No. 04-0783. The intensity ratio between the (111) and (200) peaks is 4.28 (much higher than the theoretical value of 2.5), which indicates that AgNWs are preferentially grown in the (111) crystal plane [45], which is consistent with the analysis of the growth mechanism of AgNNs. In addition, there are AgCl impurity peaks at 27.5°, 31.8° and 45.6°, which may be due to the incomplete reaction.

To verify the in situ growth of AgNWs on polyhedral particles to form AgNNs, the UV-Vis spectra of AgNWs and AgNNs-24h were analyzed under the same conditions. Figure 5d shows the Uv-Vis of AgNWs, and two peaks can be seen near 348 nm and 376 nm, which are caused by the longitudinal surface plasmon resonance (SPR) and transverse SPR of AgNWs, proving that the product contained a large amount of AgNWs [46]. Figure 5e shows the Uv-Vis of AgNNs-24 h at 350–700 nm. It can be seen that 18 resonance absorption peaks with uniform distribution and an absorbance difference less than 0.05 appear between 320 and 700 nm. With the increased incident light wavelength, the SPR effect was weakened, and the spectrum showed a decreasing curve with slight fluctuation. This is significantly different from the spectral absorption of AgNWs. The SPR peaks are caused by the mutual vibration of the free electrons resonating with the light waves, which depend on the morphology (size, shape) and refractive index of the nanometallic particles [47,48]. The appearance of complex absorption peaks of AgNNs indicates that the polyhedral particles obtained after etching and the AgNWs grown in situ differ in size and morphology and have a multi-level resonance structure, which is consistent with the structure of AgNNs observed in SEM, demonstrating the prevalence of the network structure in the samples. In addition, the absorbance of AgNNs in the near-infrared band of 800–1300 nm was also detected. As shown in Figure 5f, AgNNs-24 h had a stronger spectral absorption than the Uv-Vis band in the full near-infrared band. Resonance peaks appear in the range of 980–1160 nm and 1200–1300 nm, indicating the presence of SPR phenomena [49,50]. Research has shown that only very few materials can achieve near-infrared band absorption [51]; however, the AgNNs can achieve full near-infrared band absorption, and have great potential in the applications of Raman spectrum recognition, catalysis, sensing, medicine and other fields [52].

To evaluate the conductivity of AgNNs, the AgNNs were coated on the 4 × 15 × 0.5 mm single-crystal quartz substrate and 18 measurements were taken using the four-probe resistance meter. The measurement results show that the square resistance of the materials was between 0.13 Ω∙sq^−1^ and 0.2 Ω∙sq^−1^, with an average value of 0.15 Ω∙sq^−1^, which was 0.2 Ω∙sq^−1^ lower than the square resistance of AgNWs, which was 0.35 Ω∙sq^−1^, indicating that the conductivity of the three-dimensional conductive network formed by nano silver is better than that of AgNWs.

#### 2.2.3. Extensibility of Flexible Stretchable Conductive Film of AgNNs

NaAlg is a natural polysaccharide with good biocompatibility and biodegradability, in which the negatively charged carboxylate can form a strong electrostatic force with the Ca^+^ ions in the CaCl_2_ solution to form a cross-linked network, which is a matrix material used for the preparation of conductive fabrics with high flexibility and conductivity [53,54]. Using PDMS, paper and non-woven fabric as substrate materials, flexible conductive films were prepared using the rod coating method. At the same time, five polyester materials with different weaving processes were selected to prepare conductive fabrics using the impregnation method, and the conductivity and extensibility were tested, respectively.

The four-probe resistance tester was used to measure the electrical properties of the flexible conductive film based on AgNNs. The flexibility of the conductive film was evaluated by observing the change in the square resistance of the conductive film when it was bent many times. Figure 6a shows the object before and after the preparation of conductive films and conductive fabrics from different substrate materials, all of which appear yellow-brown with rough surfaces after treatment. The initial square resistances of PDMS, paper and non-woven AgNNs conductive films prepared using the rod method without bending were 0.56 Ω∙sq^−1^, 0.36 Ω∙sq^−1^ and 0.37 Ω∙sq^−1^, respectively. As shown in Figure 6b, the square resistance of PDMS-based conductive film had a relatively small increase within 50 times of bending. After 50 times of bending, the square resistance rose linearly with the increase in bending times, and rose to 8.7 Ω∙sq^−1^ after 500 times of bending. This was mainly caused by the poor adhesion between PDMS and the conductive materials during the bending process. After 500 times of bending, the square resistances of the paper-based and non-woven conductive films based on the AgNNs were 0.42 Ω∙sq^−1^ and 0.45 Ω∙sq^−1^, respectively, which are almost the same as those before bending, indicating that the conductive films prepared by AgNNs had good extensibility without fracture of the substrate. The average square resistances of conductive fabrics with five different weaving processes—small void, large void, coarse jacquard, pinstripe and rib pattern—were 0.27 Ω∙sq^−1^, 0.4 Ω∙sq^−1^, 0.16 Ω∙sq^−1^, 0.18 Ω∙sq^−1^ and 0.23 Ω∙sq^−1^, respectively. As shown in Figure 6c, the square resistance of the five conductive fabrics remained almost unchanged before reaching the ultimate tensile rate, and the square resistance increased sharply when the ultimate tensile rate was reached.

The ultimate tensile rates of the five conductive fabrics were 30.4%, 30.2%, 52.2%, 43.5% and 30.2%, respectively. The coarse jacquard polyester material is composed of warp and weft interlaced. The fabric is relatively compact, which can provide a good carrier for conductive materials. Therefore, its original square resistance was the lowest, and its tensile property and conductivity were the best. Table 1 demonstrates the performance of conductive fabrics prepared with AgNNs as the conductive material compared to other stretchable electrodes prepared in the literature with different processes for AgNWs. It can be seen that the conductive fabrics prepared with AgNNs as the conductive material and coarse jacquard polyester material as the substrate have a low square resistance, high conductivity, high stretchability and good durability, and therefore AgNNs have a vast application prospect as a conductive material. Table 1 shows the performance comparison of conductive fabrics prepared with AgNNs as conductive materials with stretchable electrodes prepared with AgNWs treated by different processes in other literature. It can be seen that conductive fabrics prepared using AgNNs as conductive materials and coarse jacquard polyester materials as substrates have a low square resistance, strong conductivity, high stretchability and good durability. Therefore, AgNNs as conductive materials have broad application prospects.

## 3. Materials and Methods

### 3.1. Preliminary Preparation

Silver nanoparticles (AgNPs) (99.95%, particle size 50 nm) (Zhongkedetong Technology Co., Ltd., Beijing, China), acetone (C_3_H_6_O) (AR, Macklin, Shanghai, China), propylene glycol (C_3_H_8_O_2_) (AR, Macklin), ethylene glycol ((CH_2_OH)_2_) (AR, Macklin), silver nitrate (AgNO_3_) (AR, Shanghai Aladdin Biochemical Technology Co., Ltd., Shanghai, China), polyvinylpyrrolidone (PVP) (average Mw of 40,000) (Sinopharm Chemical Reagent Co., Ltd., Shanghai, China), potassium bromide (KBr) (SP, Macklin), silver chloride (AgCl) (AR, Macklin), calcium chloride (CaCl_2_) (AR, Macklin), ethanol (Pharmaceutical grade, Macklin) and deionized water (electrophoretic grade, self-made) were used.

Preparation of AgNPs: weigh 10 g of AgNPs (99.95%, particle size 50 nm) into acetone solution, stir magnetically for 12 h and then put into a blast dryer for drying to obtain treated AgNPs.

### 3.2. Synthesis of AgNNs

The synthesis of AgNNs is shown in Figure 7. A diol solution was prepared by mixing propylene glycol and ethylene glycol in a 2:1 ratio. A total of 3.4 g of AgNO_3_ was added to 100 mL of diol solution and stirred until dissolved to obtain AgNO_3_ solution. A total of 3.35 g of PVP was weighed, added to 200 mL of diol solution and stirred magnetically until dissolving completely. Then, 0.075 g of KBr, 0.255 g of AgCl and 2 g of treated AgNPs were added and magnetically stirred at 160 °C for 24 h. The AgNO_3_ solution was added dropwise to the reaction solution using a constant flow pump at 50 rpm and stirred continuously for 5 h. After cooling to room temperature, the acetone solution was added and left to stand for 3 h and centrifuged for 20 min to obtain the AgNNs dispersion.

### 3.3. Preparation of AgNNs Flexible Stretchable Conductive Films and Conductive Fabrics

The synthesis of the AgNNs flexible stretchable conductive films and conductive fabrics is shown in Figure 8.

Preparation of flexible stretchable conductive films by rod coating method: PDMS, fibrous paper and non-woven fabrics were cut to 20 × 10 mm and ultrasonically cleaned with methanol, acetone, ethanol and deionized water for 20 min, respectively. Then, they were dried at 70 °C. A total of 0.1 g of CaCl_2_ was dissolved in 30 mL of deionized water, and then 1 g of sodium alginate (NaAlg) was taken, dissolved in CaCl_2_ solution and stirred rapidly at room temperature until the solution was viscous and transparent. Then, the solution was sealed and stood for 12 h to obtain NaAlg solution. The NaAlg sodium rod was coated on PDMS, fiber paper and non-woven fabric, and then the AgNNs dispersion was evenly coated on the three substrate materials and finally placed in a blast dryer at 60 °C. After drying, the flexible stretchable conductive films of PDMS, fiber paper and non-woven fabrics were prepared.

Preparation of conductive fabrics by impregnation method: polyester fabrics of five different weaves were cut to 23 × 10 mm, ultrasonically cleaned with acetone, methanol, ethanol and deionized water for 20 min, respectively, and dried at 60 °C. The fabrics were repeatedly lifted and soaked in 10 mg/mL mesh AgNNs dispersion. After drying at 75°C, the polyester conductive fabric was obtained.

### 3.4. Characterization

The morphology of AgNNs was characterized with a scanning electron microscope (SEM, EVO-18) at an accelerating voltage of 5 kV. The absorption spectrum of the material was measured using an ultraviolet–visible spectrophotometer (Uv-Vis, Trousse S-2600), in the wavelength range 200–1300 nm. Square resistance of AgNNs flexible stretchable conductive films and conductive fabrics was measured by the precision square resistance tester (HPS2526) and a standard four-point probe method (Four-Probe Tech.), with the probe spacing set at 2 mm. The crystal structure of AgNNs was characterized using an X-ray diffractometer (XRD, Bruker D8).

## 4. Conclusions

In this paper, AgNNs with three-dimensional conductive networks were prepared by combining the anisotropic polyhedral particles with the AgNWs growth process to achieve the anisotropic in situ growth of AgNWs on and between polyhedral particles. AgNWs were connected through polyhedral particles, which greatly improves the utilization rate of AgNWs. At the same time, it did not need to consider the secondary treatment of nodes. The material was not limited by the substrate material and can be used to prepare conductive films and conductive fabrics based on AgNNs using the rod coating or impregnation method. Among the conductive fabrics prepared with polyester fabrics with five different weaving processes as the flexible stretchable substrate, the coarse jacquard woven conductive fabric had the lowest square resistance, which was 0.16 Ω∙sq^−1^, and the square resistance remained almost unchanged within the range of 52.2%.

Therefore, AgNNs can effectively improve the conductivity and extensibility of the electrode and have broad application potential in fields such as flexible sensing, flexible display, catalysis, imaging, surface-enhanced Raman scattering (SERS) and biomedicine.

## Figures and Tables

**Figure 1 ijms-24-09319-f001:**
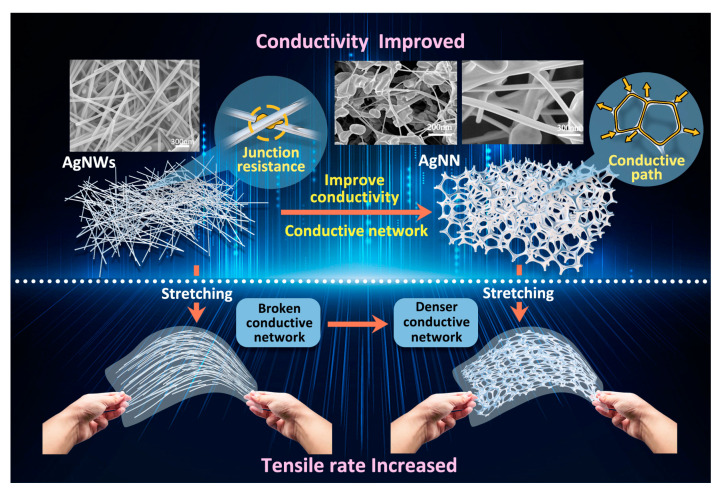
Schemes follow the same formatting. Schematic of conductivity and extensibility enhancement mechanism of flexible stretchable conductive film of silver nanonets (AgNNs).

**Figure 2 ijms-24-09319-f002:**
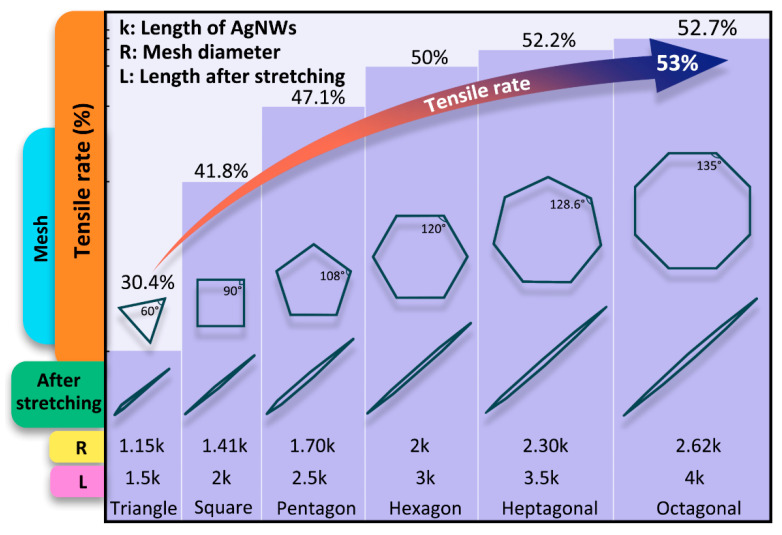
Schematic of stretching of different types of mesh.

**Figure 3 ijms-24-09319-f003:**
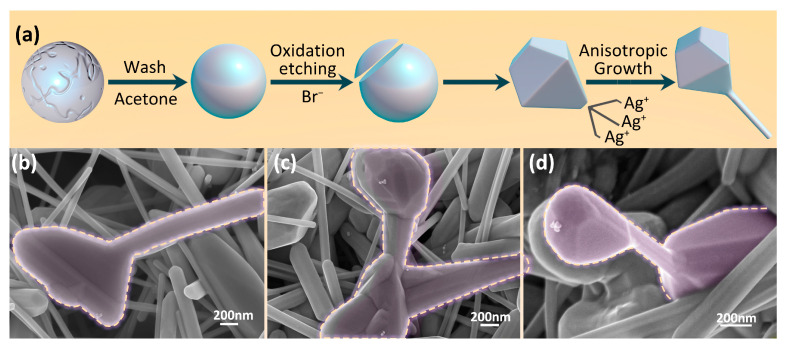
(**a**) In situ growth mechanism of AgNWs on the polyhedral particles; (**b**–**d**) SEM image of in situ growth of some AgNWs on AgNPs after adding treated AgNPs to the reaction system.

**Figure 4 ijms-24-09319-f004:**
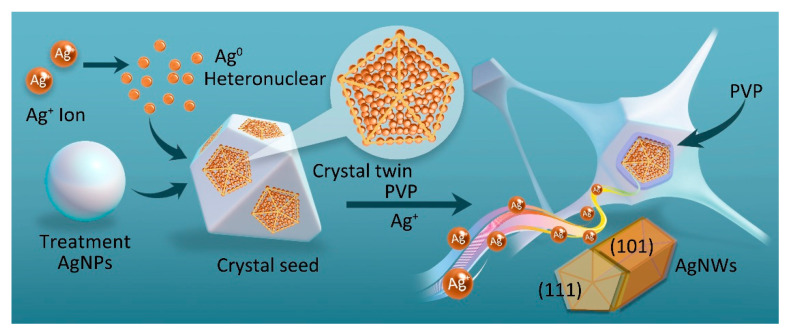
In situ growth mechanism of AgNWs on polyhedral particles.

**Figure 5 ijms-24-09319-f005:**
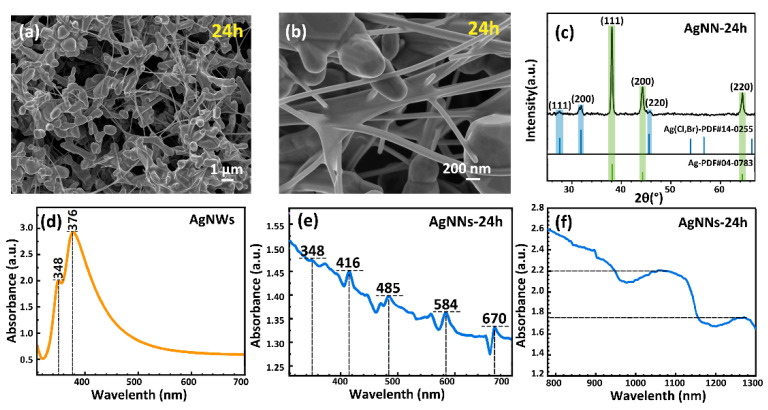
(**a**,**b**) SEM image of AgNNs-24 h (the reaction time is 24 h); (**c**) XRD pattern of AgNNs; (**d**) UV-Vis spectra of AgNWs at 350–700 nm; (**e**) UV-Vis spectra of AgNNs-24 h at 350–700 nm; (**f**) near-infrared spectra of AgNNs-24 h at 800–1300 nm.

**Figure 6 ijms-24-09319-f006:**
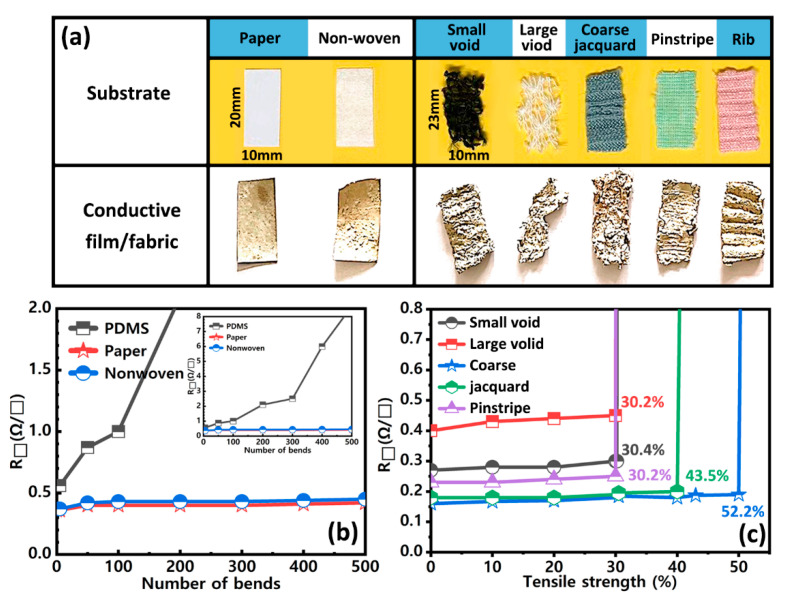
(**a**) Object of flexible substrate, conductive film and five kinds of conductive fabrics (small void, large void, coarse jacquard, pinstripe and rib pattern); square resistance results regarding (**b**) bending of PDMS, paper and non-woven AgNNs conductive films; (**c**) tensile test of five kinds of polyester conductive fabrics.

**Figure 7 ijms-24-09319-f007:**
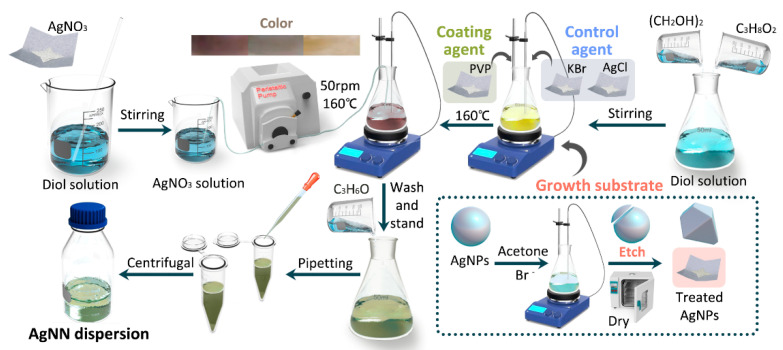
Synthesis of the AgNNs.

**Figure 8 ijms-24-09319-f008:**
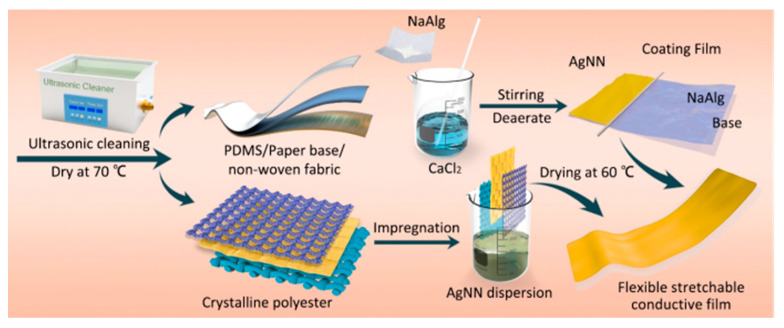
Synthesis of the AgNNs flexible stretchable conductive films and conductive fabrics.

**Table 1 ijms-24-09319-t001:** Comparison of the performance of stretchable electrodes prepared using different silver nanowires (AgNWs) treatment methods.

Treatment of AgNWs	Square Resistance (Ω∙sq^−1^)	Tensile Rate	Ref.
Brush painting	19.7	When the tensile rate is 20%, the tensile direction and brushing direction of the sample do not affect the change in resistance.	[32]
Situ growth	9	When the tensile rate is 100%, the square resistance of the conductive fabric increases from 9 Ω/sq to 12 Ω/sq.	[55]
Spray-coating	11	With the increase in tensile strain (40%) and the number of tensile cycles, the resistance of the plate increases moderately by approximately two times.	[56]
Meyer rod coating	30	When the tensile rate is 20%, there is little change in resistance	[57]
Impregnation	0.16	Before reaching the ultimate tensile ratio of 52.2%, there is little change in the square resistance of the conductive fabric	This work

## Data Availability

Not applicable.

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
