# Peer review of "In Situ Silver Nanonets for Flexible Stretchable Electrodes"

_ijms, 2023, doi:10.3390/ijms24119319_

Round 1

Reviewer 1 Report

The subject treated in the manuscript is interesting and relevant. Numerous papers deal with the study and application of flexible silver nano nets. However, my general impression is that the manuscript, although having potential, is inaccurately and vaguely written. Therefore, I feel that the manuscript needs to be significantly improved before it can be considered for publication.

The original contribution of this work is not clearly stated and should be clearly described. Especially in relation to the recent work of the research group.

The manuscript contains numerous unsubstantiated claims. For example:
Line 83: The authors claim that the ultimate tensile strength of meshes with different shapes was calculated, but no calculation can be found in the manuscript. Line 88: The authors claim, “It is found that the AgNNs have strong plasmon resonance effect in the near-infrared band.“  It is true that the spectrum in the near-infrared region (Fig. 5e) has some structure, but there is no evidence to support the claim that it is a plasmonic resonance.

In most cases, what is seen in the images is not adequately described in either the caption or the text. These descriptions should be written.

The purpose of some parts of the text is not clear. For example, the text in lines 191-201.

The structure of the manuscript is not appropriate, an should be corrected. The Materials and Methods section should come before the Results and Discussion. Large portions of the Results and Discussion section should be moved to the Materials and Methods section. The equipment used for the measurements should be described.
Mišljenja sam da je potrebno prikazati XRD spektar

I believe that it is necessary to show the XRD spectrum of the synthesized AgNNs.

The Results and Discussion section should be comprehensively improved. For example, the UV-Vis spectra of AgNNs-24 h differ significantly from similar spectra in the literature. They show a series of sharp maxima (Figure 5c) without a characteristic broad maximum just below 400 nm. How can this be explained? The purpose of plotting the curves to fit the fractional peaks in Figure 5e is not explained and should be discussed and clarified.

All abbreviations should be explained when they first appear in the text.

The terminology does not seem appropriate. For example, the term "ultimate tensile stress" used in the manuscript does not seem appropriate. Consider using the term "ultimate tensile strain" instead. Also, consider using the term "sheet resistance" instead of the term "quadratic resistance".

Author Response

Dear Reviewer,

We would like to express our thanks to the Reviewers for their instructive comments concerning our manuscript entitled ‘In-situ silver nanonets for flexible stretchable electrodes’(ijms-2370705). We have studied the comments carefully, and then revised the whole manuscript. The detail revisions/explanation corresponding to comment are shown in the attached file.

Qingwei Liao

Reviewer 2 Report

The article "In-situ silver nanonets for flexible stretchable electrodes" describes the obtaining of a silver net and its possible uses in obtaining conductive flexible films. It is a valuable study but unfortunately is plagued by some grave problems:

The English language needs some polishing for style and typos (e.g. 3.1. Subsubsection; “AgNPs treated with magnetic force”; “5 Patents”should be deleted; “Under The four”)

Source of AgNPs? If they are synthetized by authors, the employed method must be given.

The methods section must give relevant information about methods, equipment and parameters.

Which was the bending angle for the samples with AgNNs? 45o, 180o?

Try to avoid the expression like slightly, a little bit. Data analysis should be rigorous in research. (e.g. rows 274-276 are vague with no hope that other scientist could replicate the experiment)

In Figure 7 AgCl2 must be corrected.

What is “sodium alginate solution rod” that was coated? From Figure 8 looks like there was an alginate film that was used as a base. Role of CaCl2 is to reticulate the sodium alginate by replacing sodium with calcium ions and obtain an egg-box structure (see and cite doi: 10.3390/pharmaceutics13071020). Why authors have added CaCl2 before film making?

Use uniform notation for measurement units (now for litre are used both l and L like at rows 297 and 308 but also elsewhere across the manuscript). Personally, I would recommend the use of L. Same is true for UV-Vis spectrum (Uv-vis at row 201 for example).

Authors can use the term nanoparticles if the Ag particles are smaller than 100 nm. As they are smaller than 500 nm (and much larger than 200 nm as seen in Figure 3) the correct term is particles.

In the Figure 3 d-f, the purple highlight is strange and I would advise the authors to use other method to highlight the zone in SEM images, for example border dotting. This way nothing gets obscured.

A preferential growth direction should be easily detectable by XRD analysis (which should be added any way as a cheap effective analysis to identify the existence and type of Ag crystals).

The proposed mechanism must be supported by references (e.g. “Oswald” maturation – the proper spelling being Ostwald anyway). The following literature can help authors to prove their manuscript: doi 10.1155/2020/6651207; doi: 10.3390/ijms23115982 (Ostwald maturation and LSPR position in UV-Vis spectra vs type of nanoparticle).

In fact, many sentences need supporting references: “Considering that the added AgNPs have large surface energy [REF], it needs a longer reaction time” [REF]. “When the particle symmetry decreases, the number of resonances will increase significantly” [REF]. “the resonance absorption peak will be redshifted and broadened” [REF] etc.

Abstract should be checked and revised carefully by briefly introducing the work plan and key findings.

Abstracts should highlight the innovation of the article, as often abstract section is presented separately in search engines, it must be able to stand alone as an informative piece. The conclusion should reflect the heuristic of the study and novelty.

Please use some available English services or ask a proficient English colleague for help with the manuscript. Some sentences are strange, with improper words used.

Author Response

(The authors gave the same response as above.)

Reviewer 3 Report

Review of the manuscript entitled “In-situ silver nanonets for flexible stretchable electrodes” for MDPI International Journal of Molecular Sciences.

In the submitted manuscript the authors present the fabrication and characterization of silver nanonets based on the anisotropic growth of silver nanowires on etched silver nanoparticles. These nanonets are characterized using SEM microscopy, optically and electrically. The nanonets are shown to be bendable without significant loss to their properties up to a certain point. Moreover, these nanonets are grown on a variety of materials such as paper and fabrics, increasing their potential use in a variety of applications.

The subject and the presented results are very interesting however I believe that the manuscript would benefit from the clarification of a few points:

1)      The English in the manuscript should be improved. The language used in many places makes for long and awkward sentences and is sometimes difficult to follow. I suggest reediting the manuscript.

2)      The authors use the acronyms NWs, NN and NP often in the manuscript. However only the first two are defined within the text and only in the abstract. I believe that the third one should be defined, as well, and in the main text.

3)      The electrical measurement should be more explicitly described. The authors mention that they use a four-point probe method but they do not mention any details. For example, what is the size of the measured nanonet, what surface is it on, what is the distance of their probes, what type of probes did they use? These questions should also be answered in the case of the measurement of the sheet resistance while bending the nanonets.

4)      It would be helpful to include a direct comparison between sheet resistance and bendability between the presented results and the state-of-the-art in the literature. For example, in the introduction it is mentioned that other methods for creating a stronger bond between silver NWs in NW mats are more complicated and costly and that the presented method offers a much simpler road to creating bendable nanonets. There is no discussion on how the presented method compares to these more complicated methods in terms of sheet resistance results.

The English in the manuscript should be improved. The language used in many places makes for long and awkward sentences and is sometimes difficult to follow. I suggest reediting the manuscript.

Author Response

(The authors gave the same response as above.)

Round 2

Reviewer 1 Report

The authors have responded to my comments and have improved the manuscript considerably, so I propose to publish the paper in its present form.

Reviewer 2 Report

The authors have responded to my comments and have addressed my concerns, substantially improving the manuscript, therefore, I suggest publishing the paper in the current form.